# Modeling Deep Temporal Dependencies with Recurrent "Grammar Cells"

**Vincent Michalski**
Goethe University Frankfurt, Germany
vmichals@rz.uni-frankfurt.de

**Roland Memisevic**
University of Montreal, Canada
roland.memisevic@umontreal.ca

**Kishore Konda**
Goethe University Frankfurt, Germany
konda.kishorereddy@gmail.com

## Abstract

We propose modeling time series by representing the transformations that take a frame at time t to a frame at time t+1. To this end we show how a bi-linear model of transformations, such as a gated autoencoder, can be turned into a recurrent network, by training it to predict future frames from the current one and the inferred transformation using backprop-through-time. We also show how stacking multiple layers of gating units in a recurrent pyramid makes it possible to represent the "syntax" of complicated time series, and that it can outperform standard recurrent neural networks in terms of prediction accuracy on a variety of tasks.

## 1 Introduction

The predominant paradigm of modeling time series is based on state-space models, in which a hidden state evolves according to some predefined dynamical law, and an observation model maps the state to the dataspace. In this work, we explore an alternative approach to modeling time series, where learning amounts to finding an explicit representation of the *transformation* that takes an observation at time $t$ to the observation at time $t + 1$.

Modeling a sequence in terms of transformations makes it very easy to exploit redundancies that would be hard to capture otherwise. For example, very little information is needed to specify an element of the signal class *sine-wave*, if it is represented in terms of a linear mapping that takes a snippet of signal to the next snippet: given an initial "seed"-frame, any two sine-waves differ only by the amount of phase shift that the linear transformation has to repeatedly apply at each time step.

In order to model a signal as a sequence of transformations, it is necessary to make transformations "first-class objects", that can be passed around and picked up by higher layers in the network. To this end, we use bilinear models (e.g. [1, 2, 3]) which use multiplicative interactions to extract transformations from pairs of observations. We show that deep learning which is proven to be effective in learning structural hierarchies can also learn to capture hierarchies of relations or transformations. A deep model can be built by stacking multiple layers of the transformation model, so that higher layers capture higher-oder transformations (that is, transformations between transformations). To be able to model multiple steps of a time-series, we propose a training scheme called *predictive training*: after computing a deep representation of the dynamics from the first frames of a time series, the model predicts future frames by repeatedly applying the transformations passed down by higher layers, assuming constancy of the transformation in the top-most layer. Derivatives are computed using back-prop through time (BPTT) [4]. We shall refer to this model as a predictive gating pyramid (PGP) in the following.

Since hidden units at each layer encode transformations, not content of their inputs, they capture only structural dependencies and we refer to them as "grammar cells."[1] The model can also be viewed as a higher-order partial difference equation whose parameters are estimated from data. Generating from the model amounts to providing boundary conditions in the form of seed-frames, whose number corresponds to the number of layers (the order of the difference equation). We demonstrate that a two-layer model is already surprisingly effective at capturing whole classes of complicated time series, including frequency-modulated sine-waves (also known as "chirps") which we found hard to represent using standard recurrent networks.

## 1.1 Related Work

LSTM units [5] also use multiplicative interactions, in conjunction with self-connections of weight 1, to model long-term dependencies and to avoid vanishing gradients problems [6]. Instead of constant self-connections, the lower-layer units in our model can represent long-term structure by using dynamically changing *orthogonal transformations* as we shall show. Other related work includes [7], where multiplicative interactions are used to let inputs modulate connections between successive hidden states of a recurrent neural network (RNN), with application to modeling text. Our model also bears some similarity to [3] who model MOCAP data using a three-way Restricted Boltzmann Machine, where a second layer of hidden units can be used to model more "abstract" features of the time series. In contrast to that work, our higher-order units which are bi-linear too, are used to explicitly model higher-order transformations. More importantly, we use predictive training using backprop through time for our model, which is crucial for achieving good performance as we show in our experiments. Other approaches to sequence modeling include [8], who compress sequences using a two-layer RNN, where the second layer predicts residuals, which the first layer fails to predict well. In our model, compression amounts to exploiting redundancies in the relations between successive sequence elements. In contrast to [9] who introduce a recursive bi-linear autoencoder for modeling language, our model is recurrent and trained to predict, not reconstruct. The model by [10] is similar to our model in that it learns the dynamics of sequences, but assumes a simple autoregressive, rather than deep, compositional dependence, on the past. An early version of our work is described in [11].

Our work is also loosely related to sequence based invariance [12] and slow feature analysis [13], because hidden units are designed to extract structure that is invariant in time. In contrast to that work, our multi-layer models assume *higher-order invariances*, that is, invariance of velocity in the case of one hidden layer, of acceleration in the case of two, of jerk (the rate of change of acceleration) in the case of three, etc.

## 2 Background on Relational Feature Learning

In order to learn transformation features, $\mathbf{m}$, that represent the relationship between two observations $\mathbf{x}^{(1)}$ and $\mathbf{x}^{(2)}$ it is necessary to learn a basis that can represent the correlation structure across the observations. In a time series, knowledge of one frame, $\mathbf{x}^{(1)}$, typically highly constrains the distribution over possible next frames, $\mathbf{x}^{(2)}$. This suggests modeling $\mathbf{x}^{(2)}$ using a feature learning model whose parameters are a function of $\mathbf{x}^{(1)}$ [14], giving rise to bi-linear models of transformations, such as the Gated Boltzmann Machine [15, 3], Gated Autoencoder [16], and similar models (see [14] for an overview). Formally, bi-linear models learn to represent a linear transformation, $\mathbf{L}$, between two observations $\mathbf{x}^{(1)}$ and $\mathbf{x}^{(2)}$, where

$$\mathbf{x}^{(2)} = \mathbf{L}\mathbf{x}^{(1)}. \tag{1}$$

Bi-linear models encode the transformation in a layer of *mapping units* that get tuned to rotation angles in the invariant subspaces of the transformation class [14]. We shall focus on the gated autoencoder (GAE) in the following but our description could be easily adapted to other bi-linear models. Formally, the response of a layer of mapping units in the GAE takes the form[2]

$$\mathbf{m} = \sigma\big(\mathbf{W}(\mathbf{U}\mathbf{x}^{(1)} \cdot \mathbf{V}\mathbf{x}^{(2)})\big). \tag{2}$$

where $\mathbf{U}, \mathbf{V}$ and $\mathbf{W}$ are parameter matrices, $\cdot$ denotes elementwise multiplication, and $\sigma$ is an elementwise non-linearity, such as the logistic sigmoid. Given mapping unit activations, $\mathbf{m}$, and the first observation, $\mathbf{x}^{(1)}$, the second observation can be reconstructed using

$$\tilde{\mathbf{x}}^{(2)} = \mathbf{V}^{\mathrm{T}}(\mathbf{U}\mathbf{x}^{(1)} \cdot \mathbf{W}^{\mathrm{T}}\mathbf{m}) \tag{3}$$

which amounts to applying the transformation encoded in $\mathbf{m}$ to $\mathbf{x}^{(1)}$ [16]. As the model is symmetric, the reconstruction of the first observation, given the second, is similarly given by

$$\tilde{\mathbf{x}}^{(1)} = \mathbf{U}^{\mathrm{T}}(\mathbf{V}\mathbf{x}^{(2)} \cdot \mathbf{W}^{\mathrm{T}}\mathbf{m}). \tag{4}$$

For training one can minimize the symmetric reconstruction error

$$\mathcal{L} = ||\mathbf{x}^{(1)} - \tilde{\mathbf{x}}^{(1)}||^2 + ||\mathbf{x}^{(2)} - \tilde{\mathbf{x}}^{(2)}||^2. \tag{5}$$

Training turns the rows of $\mathbf{U}$ and $\mathbf{V}$ into filter pairs which reside in the invariant subspaces of the transformation class on which the model was trained. After learning, each pair is tuned to a particular rotation angle in the subspace, and the components of $\mathbf{m}$ are consequently tuned to subspace rotation angles. Due to the pooling layer, $\mathbf{W}$, they are furthermore independent of the absolute angles in the subspaces [14].

## 3 Higher-Order Relational Features

Alternatively, one can think of the bilinear model as performing a first-order Taylor approximation of the input sequence, where the hidden representation models the partial first-order derivatives of the inputs with respect to time. If we assume constancy of the first-order derivatives (or higher-order derivates, as we shall discuss), the complete sequence can be encoded using information about a single frame and the derivatives. This is a very different way of addressing long-range correlations than assuming memory units that explicitly keep state [5]. Instead, here we assume that there is structure in the temporal evolution of the input stream and we focus on capturing this structure. As an intuitive example, consider a sinusoidal signal with unknown frequency and phase. The complete signal can be specified exactly and completely after having seen a few seed frames, making it possible in principle to generate the rest of the signal ad infinitum.

### 3.1 Learning of Higher-Order Relational Features

The first-order partial derivative of a multidimensional discrete-time dynamical system describes the correspondences between observations at subsequent time steps. The fact that relational feature learning applied to subsequent frames may be viewed as a way to learn these derivatives, suggests modeling higher-order derivatives with another layer of relational features.

To this end, we suggest cascading relational features in a "pyramid" as depicted in Figure 1 on the left.[3] Given a sequence of inputs $\mathbf{x}^{(t-2)}, \mathbf{x}^{(t-1)}, \mathbf{x}^{(t)}$, first-order relational features $\mathbf{m}_1^{(t-1:t)}$ describe the transformations between two subsequent inputs $\mathbf{x}^{(t-1)}$ and $\mathbf{x}^{(t)}$. Second-order relational features $\mathbf{m}_2^{(t-2:t)}$ describe correspondences between two first-order relational features $\mathbf{m}_1^{(t-2:t-1)}$ and $\mathbf{m}_1^{(t-1:t)}$, modeling the "second-order derivatives" of the signal with respect to time.

To learn the higher-order features, we can first train a bottom-layer GAE module to represent correspondences between frame pairs using filter matrices $\mathbf{U}_1, \mathbf{V}_1$ and $\mathbf{W}_1$ (the subscript index refers to the layer). From the first-layer module we can infer mappings $\mathbf{m}_1^{(t-2:t-1)}$ and $\mathbf{m}_1^{(t-1:t)}$ for overlapping input pairs $(\mathbf{x}^{(t-2)}, \mathbf{x}^{(t-1)})$ and $(\mathbf{x}^{(t-1)}, \mathbf{x}^{(t)})$, and use these as inputs to a second-layer GAE module. A second GAE can then learn to represent relations between mappings of the first-layer using parameters $\mathbf{U}_2, \mathbf{V}_2$ and $\mathbf{W}_2$.

Inference of second-order relational features amounts to computing first- and second-order mappings according to

$$\mathbf{m}_1^{(t-2:t-1)} = \sigma(\mathbf{W}_1((\mathbf{U}_1\mathbf{x}^{(t-2)}) \cdot (\mathbf{V}_1\mathbf{x}^{(t-1)}))) \tag{6}$$

$$\mathbf{m}_1^{(t-1:t)} = \sigma(\mathbf{W}_1((\mathbf{U}_1\mathbf{x}^{(t-1)}) \cdot (\mathbf{V}_1\mathbf{x}^{(t)}))) \tag{7}$$

$$\mathbf{m}_2^{(t-2:t)} = \sigma(\mathbf{W}_2((\mathbf{U}_2\mathbf{m}_1^{(t-2:t-1)}) \cdot (\mathbf{V}_2\mathbf{m}_1^{(t-1:t)}))). \tag{8}$$

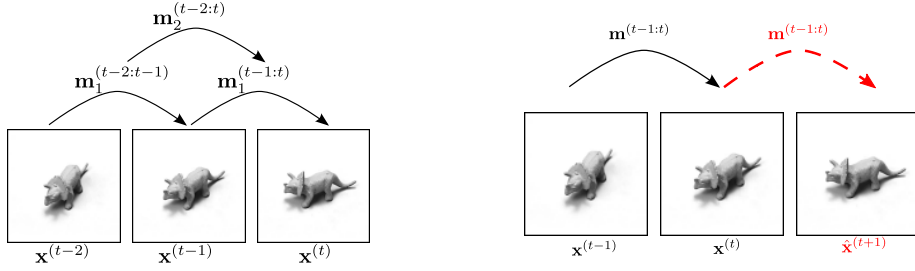

Figure 1: Left: A two-layer model encodes a sequence by assuming constant "acceleration". Right: Prediction using first-order relational features.

Like a mixture of experts, a bi-linear model represents a highly non-linear mapping from $\mathbf{x}^{(1)}$ to $\mathbf{x}^{(2)}$ as a mixture of linear (and thereby possibly orthogonal) transformations. Similar to the LSTM, this facilitates error back-propagation, because orthogonal transformations do not suffer from vanishing/exploding gradient problems. This may be viewed as a way of generalizing LSTM [5] which uses the identity matrix as the orthogonal transformation. "Grammar units" in contrast try to model long-term structure that is *dynamic* and *compositional* rather than remembering a fixed value.

Cascading GAE modules in this way can also be motivated from the view of orthogonal transformations as subspace rotations: summing over filter-response products can yield transformation detectors which are sensitive to relative angles (phases in the case of translations) and invariant to the absolute angles [14]. The relative rotation angle (or phase delta) between two projections is itself an angle, and the relation between two such angles represents an "angular acceleration" that can be picked up by another layer.

In contrast to a single-layer, two-frame model, the reconstruction error is no longer directly applicable (although a naive way to train the model would be to minimize reconstruction error for each pair of adjacent nodes in each layer). However, a natural way of training the model on sequential data is to replace the reconstruction task with the objective of *predicting* future frames as we discuss next.

## 4 Predictive Training

### 4.1 Single-Step Prediction

In the GAE model, given two frames $\mathbf{x}^{(1)}$ and $\mathbf{x}^{(2)}$ one can compute a prediction of the third frame by first inferring mappings $\mathbf{m}^{(1,2)}$ from $\mathbf{x}^{(1)}$ and $\mathbf{x}^{(2)}$ (see Equation 2) and using these to compute a prediction $\hat{\mathbf{x}}^{(3)}$ by applying the inferred transformation $\mathbf{m}^{(1,2)}$ to frame $\mathbf{x}^{(2)}$

$$\hat{\mathbf{x}}^{(3)} = \mathbf{V}^{\mathrm{T}}\big(\mathbf{U}\mathbf{x}^{(2)} \cdot \mathbf{W}^{\mathrm{T}}\mathbf{m}^{(1,2)}\big). \tag{9}$$

See Figure 1 (right side) for an outline of the prediction scheme. The prediction of $\mathbf{x}^{(3)}$ is a good prediction under the assumption that frame-to-frame transformations from $\mathbf{x}^{(1)}$ to $\mathbf{x}^{(2)}$ and from $\mathbf{x}^{(2)}$ to $\mathbf{x}^{(3)}$ are approximately the same, in other words if transformations themselves are assumed to be approximately *constant* in time. We shall show later how to relax the assumption of constancy of the transformation by adding layers to the model.

The training criterion for this *predictive gating pyramid* (PGP) is the prediction error

$$\mathcal{L} = ||\hat{\mathbf{x}}^{(3)} - \mathbf{x}^{(3)}||_2^2. \tag{10}$$

Besides allowing us to apply bilinear models to sequences, this training objective, in contrast to the reconstruction objective, can guide the mapping representation to be invariant to the content of each frame, because encoding the content of $\mathbf{x}^{(2)}$ will not help predicting $\mathbf{x}^{(3)}$ well.

### 4.2 Multi-Step Prediction and Non-Constant Transformations

We can iterate the inference-prediction process in order to look ahead more than one frame in time. To compute a prediction $\hat{\mathbf{x}}^{(4)}$ with the PGP, for example, we can infer the mappings and prediction:

$$\mathbf{m}^{(2:3)} = \sigma\big(\mathbf{W}(\mathbf{U}\mathbf{x}^{(2)} \cdot \mathbf{V}\hat{\mathbf{x}}^{(3)})\big), \qquad \hat{\mathbf{x}}^{(4)} = \mathbf{V}^{\mathrm{T}}\big(\mathbf{U}\hat{\mathbf{x}}^{(3)} \cdot \mathbf{W}^{\mathrm{T}}\mathbf{m}^{(2:3)}\big). \tag{11}$$

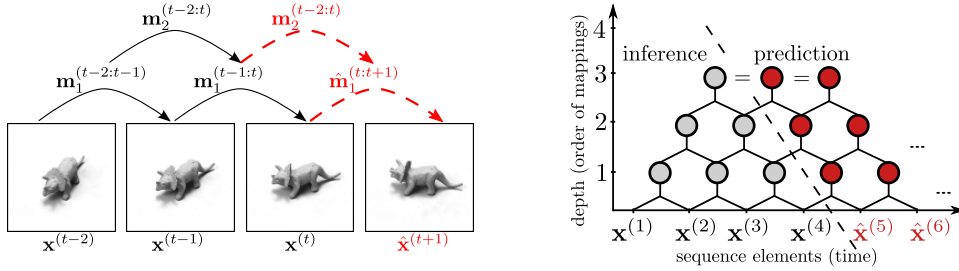

Figure 2: Left: Prediction with a 2-layer PGP. Right: Multi-step prediction with a 3-layer PGP.

Then mappings can be inferred again from $\hat{\mathbf{x}}^{(3)}$ and $\hat{\mathbf{x}}^{(4)}$ to compute a prediction of $\hat{\mathbf{x}}^{(5)}$, and so on.

When the assumption of constancy of the transformations is violated, one can use an additional layer to model how transformations themselves change over time as described in Section 3. The assumption behind the two-layer PGP is that the second-order relational structure in the sequence is constant. Under this assumption, we compute a prediction $\hat{\mathbf{x}}^{(t+1)}$ in two steps after inferring $\mathbf{m}_2^{(t-2:t)}$ according to Equation 8: First, first-order relational features describing the correspondence between $\mathbf{x}^{(t)}$ and $\mathbf{x}^{(t+1)}$ are inferred top-down as

$$\hat{\mathbf{m}}_1^{(t:t+1)} = \mathbf{V}_2^{\mathrm{T}} \big( \mathbf{U}_2 \mathbf{m}_1^{(t-1:t)} \cdot \mathbf{W}_2^{\mathrm{T}} \mathbf{m}_2^{(t-2:t)} \big), \tag{12}$$

from which we can compute $\hat{\mathbf{x}}^{(t+1)}$ as

$$\hat{\mathbf{x}}^{(t+1)} = \mathbf{V}_1^{\mathrm{T}} \big( \mathbf{U}_1 \mathbf{x}^{(t)} \cdot \mathbf{W}_1^{\mathrm{T}} \hat{\mathbf{m}}_1^{(t:t+1)} \big). \tag{13}$$

See Figure 2 (left side) for an illustration of the two-layer prediction scheme. To predict multiple steps ahead we repeat the inference-prediction process on $\mathbf{x}^{(t-1)}$, $\mathbf{x}^{(t)}$ and $\hat{\mathbf{x}}^{(t+1)}$, i.e. by appending the prediction to the sequence and increasing $t$ by one.

As outlined in Figure 2 (right side), the concept can be generalized to more than two layers by recursion to yield higher-order relational features. Weights can be shared across layers, but we used untied weights in our experiments.

To summarize, the prediction process consists in iteratively computing predictions of the next lower levels activations beginning from the top. To infer the top-level activations themselves, one needs a number of seed frames corresponding to the depth of the model. The models can be trained using BPTT to compute gradients of the $k$-step prediction error (the sum of prediction errors) with respect to the parameters. We observed that starting with few prediction steps and iteratively increasing the number of prediction steps as training progresses considerably stabilizes the learning.

## 5 Experiments

We tested and compared the models on sequences and videos with varying degrees of complexity, from synthetic constant to synthetic accelerated transformations to more complex real-world transformations. A description of the synthetic shift and rotation data sets is provided in the supplementary material.

### 5.1 Preprocessing and Initialization

For all data sets, except for chirps and bouncing balls, PCA whitening was used for dimensionality reduction, retaining around $95\%$ of the variance. The chirps-data was normalized by subtracting the mean and dividing by the standard deviation of the training set. For the multi-layer models we used greedy layerwise pretraining before predictive training. We found pretraining to be crucial for the predictive training to work well. Each layer was pretrained using a simple GAE, the first layer on input frames, the next layer on the inferred mappings. Stochastic gradient descent (SGD) with learning rate 0.001 and momentum 0.9 was used for all pretraining.

Table 1: Classification accuracies (%) on accelerated transformation data using mappings from different layers in the PGP (accuracies after pretraining shown in parentheses).

| Data set | $\mathbf{m}_1^{(1:2)}$ | $\mathbf{m}_1^{(2:3)}$ | $(\mathbf{m}_1^{(1:2)}, \mathbf{m}_1^{(2:3)})$ | $\mathbf{m}_2^{(1:3)}$ |
|---|---|---|---|---|
| AccRot | 18.1 (19.4) | 29.3 (30.9) | 74.0 (64.9) | 74.4 (53.7) |
| AccShift | 20.9 (20.6) | 34.4 (33.3) | 42.7 (38.4) | 80.6 (63.4) |

## 5.2 Comparison of Predictive and Reconstructive Training

To evaluate whether predictive training (PGP) yields better representations of transformations than training with a reconstruction objective (GAE), we first performed a classification experiment on videos showing artificially transformed natural images. $13 \times 13$ patches were cropped from the Berkeley Segmentation data set (BSDS300) [18]. Two data sets with videos featuring constant velocity shifts (ConstShift) and rotations (ConstRot) were generated. The shift vectors (for ConstShift) and rotation angles (for ConstRot) were each grouped into 8 bins to generate labels for classification.

The numbers of filter pairs and mapping units were chosen using a grid search. The setting with the best performance on the validation set was 256 filters and 256 mapping units for both training objectives on both data sets. The models were each trained for 1 000 epochs using SGD with learning rate 0.001 and momentum 0.9. Mappings of the first two inputs were used as input to a logistic regression classifier. The experiment was performed three times on both data sets. The mean accuracy (%) on ConstShift after predictive training was 79.4 compared to 76.4 after reconstructive training. For ConstRot mean accuracies were 98.2 after predictive and 97.6 after reconstructive training. This confirms that predictive training yields a more explicit representation of transformations, that is less dependent on image content, as discussed in Section 4.1.

## 5.3 Detecting Acceleration

To test the hypothesis that the PGP learns to model second-order correspondences in sequences, image sequences with accelerated shifts (AccShift) and rotations (AccRot) of natural image patches were generated. The acceleration vectors (for AccShift) and angular rotations (for AccRot) were each grouped into 8 bins to generate output labels for classification.

Numbers of filter pairs and mapping units were set to 512 and 256, respectively, after performing a grid search. After pretraining, the PGP was trained using SGD with learning rate 0.0001 and momentum 0.9, for 400 epochs on single-step prediction and then 500 epochs on two-step prediction.

After training, first- and second-layer mappings were inferred from the first three frames of the test sequences. The classification accuracies using logistic regression with second-layer mappings of the PGP ($\mathbf{m}_2^{(1:3)}$), with individual first-layer mappings ($\mathbf{m}_1^{(1:2)}$ and $\mathbf{m}_1^{(2:3)}$), and with their concatenation ($\mathbf{m}_1^{(1:2)}, \mathbf{m}_1^{(2:3)}$) as classifier inputs are compared in Table 1 for both data sets (before and after predictive finetuning). The second-layer mappings achieved a significantly higher accuracy for both data sets after predictive training. For AccRot, the concatenation of first-layer mappings performs almost as well as the second-layer mappings, which may be because rotations have fewer degrees of freedom than shifts making them easier to model. Note that the accuracy for the first layer mappings also improved with predictive finetuning.

These results show that the PGP can learn a better representation of the second-order relational structure in the data than the single-layer model. They further show that predictive training improves performances of both models and is crucial for the PGP.

## 5.4 Sequence Prediction

In these experiments we test the capability of the models to predict previously unseen sequences multiple steps into the future. This allows us to assess to what degree modeling higher order "derivatives" makes it possible to capture the temporal evolution of a signal without resorting to an explicit

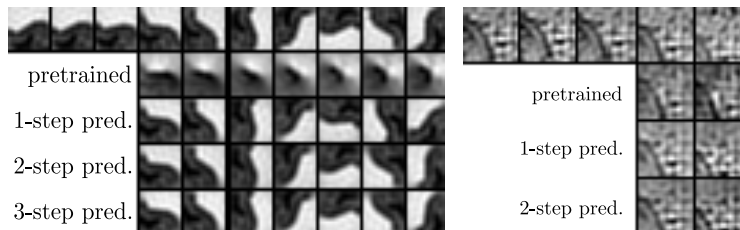

Figure 3: Multi-step predictions by the PGP trained on accelerated rotations (left) and shifts (right). From top to bottom: ground truth, predictions before and after predictive finetuning.

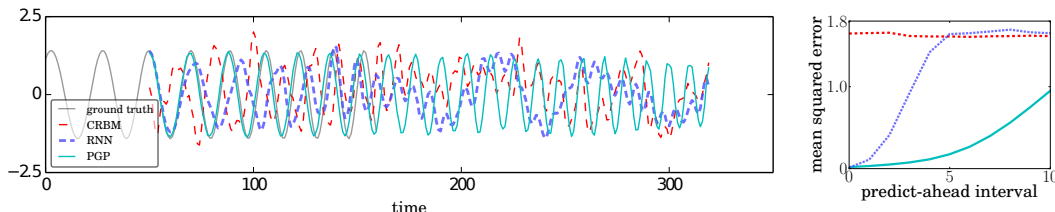

Figure 4: Left: Chirp signal and the predictions of the CRBM, RNN and PGP after seeing the first five 10-frame vectors. Right: The MSE of the three models for each step.

representation of a hidden state. Unless mentioned otherwise, the presented sequences were seeded with frames from test data (not seen during training).

**Accelerated Transformations**
Figure 3 shows predictions with the PGP on the data sets introduced in Section 5.3 after different stages of training. As can be seen in the figures, the prediction accuracy increases significantly with multi-step training.

**Chirps**
Performances of the PGP were compared with that of a standard RNN (trained with BPTT) and a CRBM (trained with contrastive divergence) [19] on a dataset containing chirps (sinusoidal waves that increase or decrease in frequency over time). Training and test set each contain $20,000$ sequences. The 160 frames of each sequence are grouped into 16 non-overlapping 10-frame windows, yielding 10-dimensional input vectors. Given the first 5 windows, the remaining 11 windows have to be predicted. Second-order mappings of the PGP are averaged for the seed windows and then held fixed for prediction. Predictions for one test sequence are shown in Figure 4 (left). Mean-squared errors (MSE) on the test set are $1.159$ for the RNN, $1.624$ for the CRBM and $0.323$ for the PGP. A plot of per-step MSEs is shown in Figure 4 (right).

**NorbVideos**
The NORBvideos data set introduced in [20] contains videos of objects from the NORB dataset [17]. The 5 frame videos each show incrementally changed viewpoints of one object. One- and two-hidden layer PGP models were trained on this data using the author's original split. Both models used 2000 features and 1000 mapping units (per layer). The performance of the one-hidden layer model stopped improving at 2000 features, while the two-hidden layer model was able to make use of the additional parameters. Two-step MSEs on test data were $448.4$ and $582.1$, respectively.

Figure 6 shows predictions made by both models. The second-order PGP generates predictions that reflect the 3-D structure in the data. In contrast to the first-order PGP, it is able to extrapolate the observed transformations.

**Bouncing Balls**
The PGP is also able to capture the highly non-linear dynamics in the bouncing balls data set[4]. The sequence shown in Figure 5 contains 56 frames, where the first 5 are from the training sequences and are used as seed for sequence generation (similar to the chirps experiment the average top-layer mapping vector for the seed frames is fixed). Note that the sequences used for training were only

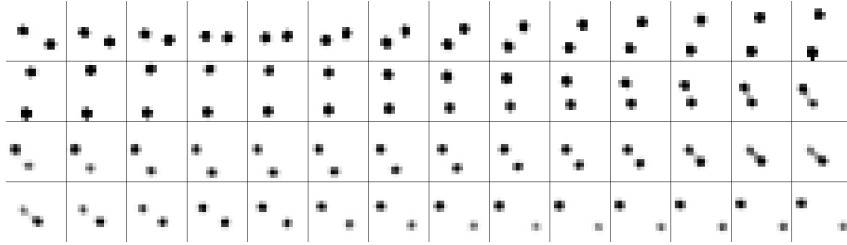

Figure 5: PGP generated sequence of bouncing balls (left-to-right, top-to-bottom).

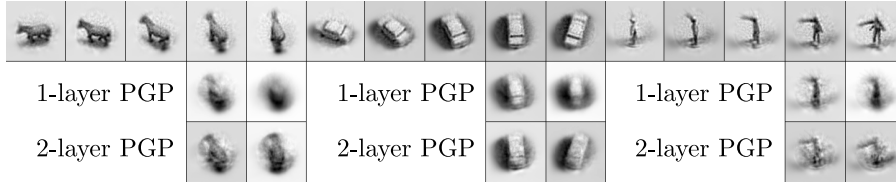

Figure 6: Two-step PGP test predictions on NORBvideos.

20 frames long. The model's predictions look qualitatively better than most published generated sequences.[5] Further results and data can be found on the project website at `http://www.ccc.cs.uni-frankfurt.de/people/vincent-michalski/grammar-cells`

# 6    Discussion

A major long-standing problem in sequence modeling is dealing with long-range correlations. It has been proposed that deep learning may help address this problem by finding representations that capture better the abstract, semantic content of the inputs [22]. In this work we propose learning representations with the explicit goal of enabling *the prediction* of the temporal evolution of the input stream multiple time steps ahead. Thus we seek a hidden representation that captures those aspects of the input data which allow us to make predictions about the future.

As we discussed, learning the long-term evolution of a sequence can be simplified by modeling it as a sequence of temporally varying orthogonal (and thus, in particular, linear) transformations. Since gating networks are like mixtures-of-experts, the PGP does model its input using a sequence of linear transformations in the lowest layer, it is thus "horizontally linear". At the same time, it is "vertically compressive", because its sigmoidal units are encouraged to compute non-linear, sparse representations, like the hidden units in any standard feed-forward neural network. From an optimization perspective this is a very sensible way to model time-series, since gradients have to be back-propagated through many more layers horizontally (in time) than vertically (through the non-linear network).

It is interesting to note that predictive training can also be viewed as an analogy making task [15]. It amounts to relating the transformation from frame $t - 1$ to $t$ with the transformation between a later pair of observations, e.g. those at time $t$ and $t + 1$. The difference is that in a genuine analogy making task, the target observation may be unrelated to the source observation pair, whereas here target and source are related. It would be interesting to apply the model to word representations, or language in general, as this is a domain where both, sequentially structured data and analogical relationships play central roles.

**Acknowledgments**

This work was supported by the German Federal Ministry of Education and Research (BMBF) in project 01GQ0841 (BFNT Frankfurt), by an NSERC Discovery grant and by a Google faculty research award.

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

## Footnotes

[1] We dedicate this paper to the venerable grandmother cell, a grandmother of the grammar cell.

[2] We are only using "factored" [15] bi-linear models in this work, but the framework presented in this work could be applied to unfactored models, too.

[3]Images taken from the NORB data set described in [17]

[4] The training and test sequences were generated using the script released with [21].

[5]compare with `http://www.cs.utoronto.ca/~ilya/pubs/2007/multilayered/index.html` and `http://www.cs.utoronto.ca/~ilya/pubs/2008/rtrbm_vid.tar.gz`.
