[Supplementary Material]

# Modeling Deep Temporal Dependencies with Recurrent "Grammar Cells" (Supplementary Material)

**Vincent Michalski**
Goethe University Frankfurt, Germany
vmichals@rz.uni-frankfurt.de

**Roland Memisevic**
University of Montreal, Canada
roland.memisevic@umontreal.ca

**Kishore Konda**
Goethe University Frankfurt, Germany
konda.kishorereddy@gmail.com

## 1  Bouncing Balls Prediction

Figure 1 shows two bouncing ball sequences [1]. generated by the predictive gating pyramid (PGP) after seeing 5 frames from the training set[2]. Note that the training sequences were generated independently and are only 20 frames long, much shorter than the generated sequences shown here.

Figure 1: Long sequence prediction of bouncing balls videos.

## 2 Chirps Prediction

Figure 2 shows eight more test cases from the chirps data set, together with predictions (beyond the ground truth horizon of 160) of the PGP, the standard recurrent neural network (RNN) and the Conditional Restricted Boltzmann Machine (CRBM) [2]. We used the Theano [3] implementations of the RNN and CRBM written by Graham Taylor, available at `http://www.uoguelph.ca/~gwtaylor/code/`.

Figure 2: Some test predictions on chirp data.

## 3 NorbVideos

Figure 3 shows left and right receptive fields of the bottom layer which developed during predictive finetuning of the PGP. Due to the large input dimensionality (frame size $96 \times 96$) and the low number of training samples a few of the filters seem to be overfitting on the training data while many others are localized Gabor-like features.

## 4 Description of the Shift and Rotation Data Sets

### 4.1 Constant-Velocity Transformations

$13 \times 13$ patches were cropped from the Berkeley Segmentation data set (BSDS300) [5]. Two data sets with videos featuring constant velocity shifts (CONSTSHIFT) and rotations (CONSTROT) were generated. Elements of the shift vectors for CONSTSHIFT were sampled uniformly from the interval $[-3, 3]$ (in pixels), and rotation angles from $(-\pi, \pi)$. Eight labels for CONSTSHIFT were assigned by partitioning shift angles into four quadrants and the shift magnitude into two bins. For CONSTROT rotation angles were divided into 8 equally-sized bins. Both data sets were partitioned into training, validation and testing set of size $100\,000$, $20\,000$ and $50\,000$, respectively.

## 4.2 Accelerated Transformations

Patches were again cropped from BSDS300 and artificially transformed with initial (angular) velocity and constant (angular) acceleration. Scalar angular accelerations were sampled uniformly from the interval $\left[-\frac{\pi}{12}, \frac{\pi}{12}\right]$ degrees. Initial angular velocities were sampled from the same interval. Angular accelerations were divided into $8$ equally sized bins. For shifts, elements of the velocity and acceleration vectors were sampled from the interval $[-3, 3]$ (in pixels). Acceleration vectors were discretized in the same way as shift vectors in CONSTSHIFT.

## Footnotes

[1] The training and test sequences were generated using the script released with [1].

[2] An animated version of these can be found in the supplementary zip archive.

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

Figure 3: Receptive fields of the PGP trained on NORBvideos [4]