[Reviews · NeurIPS 2014]

Submitted by Assigned_Reviewer_5

Summary:

The paper presents an extension of gated auto encoders to time-series data. The main idea is to use a gated auto encoder to model the time series in an autoregressive manner; predicting x_{t+1} from x_t using a gated autoencoder whose mapping unit values are initialised using a pair of contiguous datapoints. The paper introduces two interesting refinements: predictive training, and higher order relational features.

Predictive training is a training criterion suitable for time series data that is different from the criterion normally used for gated auto encoders. Predictive training tries to minimise the square error in predicting x_{t+1} given x_{t} and the value of the mapping units that optimally predict x_{t} given x_{t-1}.

Higher order relational features are an extension of predictive gating networks that relaxes the assumption of constant transformation across time. This is done by using another predictive gating network to model the progress of the mapping units (which encode the transformations), effectively adding a new layer of hidden mapping units. This scheme can be easily replicated to an arbitrary number of layers. The parameters of all k layers of mapping units can be trained jointly by using k-step-ahead predictive training, although the authors found pre training as a regular gated auto encoder to improve results.

Quality:
The paper is technically sound and claims are supported by experimental results. The ability of a 2-layer DPGM to capture accelerations is most convincingly shown in 5.4.2, but comparison with 2-layer CRBM would be fairer. Also, in the bouncing ball and NORB videos experiments quantitative comparisons between PGM, DPGM and other competing methods would have been a good addition. Due to space constraints it is not possible to show generated sequences of bouncing balls from other methods, but it could have been added in supplementary material.

Clarity:
The paper is, in general, clearly written and organised.

It is not clearly explained why the authors found pre training using GAEs “crucial” to obtain good results. Was uninitialised predictive training optimisation not able to obtain good squared prediction errors on the training set? did the resulting models not generalise to test data?

Originality:
The same problem has been addressed using similar techniques using multilayer CRBMs. However the use of gated auto encoders instead of RBMs facilitates the training and explicitly models the transformations between consecutive data points.

The authors may be interested in this other paper modelling chirp-like data:
J. Luttinen, T. Raiko, and A. Ilin.
Linear State-Space Model with Time-Varying Dynamics.
To appear in the proceedings of the European Conference on Machine Learning (ECML), September 2014.

Significance:
The results are important. The Deep PGM obtains qualitatively better results than other models and is surely of interest for the NIPS community.
Summary: The paper is an interesting extension of gated auto encoders to time-series data that introduces several refinements and obtains good qualitative results.

Submitted by Assigned_Reviewer_17

Review of submission 1062:
Modeling sequences with a predictive gating network

Summary: The authors use BPTT to predict future observations from previous observations, and build a stack of such predictors. Applications: sine waves, rotating 3D objects. The approach outperforms certain traditional RNNs.

Comments:

An interesting approach, but relations to similar previous work are missing:

1. Submission: "We show that we can build a deep model by stacking multiple layers of the transformation model, so that higher layers capture higher-oder transformations, in other words, transformations between transformations."

The first very similar hierarchy of RNN predictors was described in 1992 [17]. Each RNN predicts its next input. Each is trained by BPTT. Higher-level RNNs see only inputs that lower levels could not predict. The approach in the submission seems very similar, although it is not quite the same - what exactly are the differences?

[17] J. Schmidhuber. Learning complex, extended sequences using the principle of history compression, Neural Computation, 4(2):234-242, 1992

One difference seems to be: "after extracting the transformation from two frames in the time series, the model predicts the next frame(s) by applying the transformation passed down by a higher layer (in lower layers of the model), or by assuming constancy of the transformation through time (in the top-most layer)."

That is, unlike the apparently more general 1992 model, only 2 frames are used to predict the third? Perhaps that's why the authors are drawing this analogy: "One may think of the bilinear model as performing a first-order Taylor approximation of the input sequence, where the hidden representation models the partial first-order derivatives of the inputs with respect to time."

Another difference seems to be that the 1992 hierarchy compresses the observation sequence in unsupervised fashion, while the system of the authors does not.

"More importantly, we use predictive training using backprop through time for our model, and show that this is crucial for achieving good prediction performance."

This, however, also applies to the deep RNN hierarchy [17].

The relation to the previous work should be made very clear!

2. Submission says: "Multiplicative interactions (bilinear models) were recently shown to be useful in recurrent neural networks by [4]."

In fact, ref [8] of the authors used multiplicative interactions for RNNs much earlier, with great success.

3. Experiments:

Chirp signal: "The performances of the DPGM were compared with that of a standard RNN (trained with BPTT) and a CRBM (trained with contrastive divergence) [12] on a dataset containing chirps (sinusoidal waves that increase or decrease in frequency over time)."

It is well known that standard RNNs trained by BPTT don't do well on (superimposed) sine waves, e.g., [18]. But there are RNN training algorithms where the connections to output units are trained by the pseudoinverse, and connections to hidden units are evolved to minimize the residual error. This works very well on (even superimposed) sine waves [18]. To compare to the state of the art, one should test against such systems.

[18] J. Schmidhuber, D. Wierstra, M. Gagliolo, F. Gomez. Training Recurrent Networks by Evolino. Neural Computation, 19(3): 757-779, 2007

The other experiments are interesting.

4. Discussion

"A major long-standing problem in sequence modeling is how to deal with long range correlations."

Here one should cite the work [19] that described and analyzed this problem first:

[19] S. Hochreiter. Diploma thesis, TUM, 1991.

5. General comment: Interesting and publishable, but only provided the missing relations to similar previous work are made very clear.

Summary: Interesting and potentially publishable, but only provided the missing relations to similar previous work are made very clear.

Submitted by Assigned_Reviewer_18

Paper introduces a predictive model for sequences. One layer deep model extracts motion features from two consecutive frames using multiplicative interactions and then uses the motion features and the last frame to predict then next frame. Two layer deep model in addition uses two consecutive motion features to predict next motion feature and uses that for next frame prediction. The algorithm is tested on translated and rotated image patches (with and without acceleration), chirps, bouncing balls and norb videos. It is a good idea and should be published though experiments could be more significant. First, there could be some standard benchmark temporal dataset to compare to other methods and second, the comparison to state of the art rnn - the LSTM - is missing. Also unfortunately the network needs two stages of pre-training.
Summary: Good idea, though it would be good to know how well it works on real life sequences and how it compares to LSTM.
Author Feedback
Author rebuttal: We thank the reviewers for their comments and appreciate the constructive and positive tone used by all three reviewers.

---
Assigned_Reviewer_17:
Thank you for the references. We will add all of them as well as a discussion about their differences. More specifically,
- "only 2 frames are used to predict the third?"
Yes, this is true at each layer. And it thus amounts to using accordingly more frames for predicting the observations in the lowest layer in multi-layer models (similar to [17]).
- "...the 1992 hierarchy compresses the observation sequence in unsupervised fashion, while the system of the authors does not."
Yes, this is correct. Any "compression" in our model necessarily follows from the way that a part of the multidimensional input sequences relates to another part of the sequence. We will mention this in more detail.
- "This [backprop through time], however, also applies to the deep RNN hierarchy [17]."
Correct, we will mention this. (Our current comment in the submission was targeted at similar three-way models like [3] and [12] which had not used bptt)
- "ref [8] of the authors used multiplicative interactions for RNNs much earlier, with great success"
Yes, this is true. We will point this out.
- it is well known that standard RNNs trained by BPTT don't do well on (superimposed) sine waves, e.g., [18].
We were able to train standard RNNs on individual sine-waves but not on chirps. Since chirps cannot be modelled well by superimposing sine-waves (unless of course using a complete Fourier basis) we assume that a model doing well on superimposed sine-waves won't necessarily do well on chirps.
- one should cite the work [19]
True, we will add this reference.

Assigned_Reviewer_18:
- "there could be some standard benchmark temporal dataset to compare to other methods"
We agree it would be good if there was a standard one. Though the bouncing balls probably comes closest to a standard multivariate time-series task on which many models have been tried, and our models does much better than any model we are aware of.
- "how it compares to LSTM"
We agree a detailed comparison with LSTM will be interesting. We were however quite satisfied with the quite good performance in comparison to a wide variety of other existing models in this first paper on this model.
- the network needs two stages of pre-training.
It would be nice if we could do without any pretraining, but we found the fact that layer-by-layer pretraining is needed for good performance interesting, as this may point to a problem domain where (the existing) supervised learning methods don't seem to work at all.

Assigned_Reviewer_5:
- "it is not possible to show generated sequences of bouncing balls from other methods, but it could have been added in supplementary material"
We agree, and we will add these in the supplementary material.
- "Was uninitialised predictive training optimisation not able to obtain good squared prediction errors on the training set? did the resulting models not generalise to test data? "
Yes, it is a problem of optimisation rather than of generalisation, as uninitialised predictive training was not able to obtain a low MSE on the training set. We will clarify this point.
- "The authors may be interested in this other paper modelling chirp-like data:
J. Luttinen, T. Raiko, and A. Ilin.
Linear State-Space Model with Time-Varying Dynamics."
Thanks for the reference, which we will include (one interesting difference to that work is that a GAE itself is used recursively in our work to model how dynamics change over time, including the possibilty of stacking to obtain multi-layer models as we have shown).